# A Triple Threat: A Case Report Detailing Surgical Management for Hypertrophic Cardiomyopathy, Flail Mitral Valve and Severe Pulmonary Hypertension [note 1]

**DOI:** 10.3390/reports7040116

**Published:** 2024-12-17

**Authors:** Cass G. G. Sunga, Kai-Chun Yang, Shakirat Oyetunji, Erik R. Swenson, Kavita Khaira

**Affiliations:** 1General Internal Medicine, Department of Medicine, University of Washington, Seattle, WA 98195, USA; sungac@uw.edu; 2Division of Cardiology, Department of Medicine, University of Washington, Seattle, WA 98195, USA; kcyang@uw.edu; 3Veteran’s Health Administration Puget Sound Health Care System, Seattle, WA 98108, USAeswenson@uw.edu (E.R.S.); 4Division of Cardiothoracic Surgery, Department of Surgery, University of Washington, Seattle, WA 98195, USA; 5Division of Pulmonary, Critical Care and Sleep Medicine, Department of Medicine, University of Washington, Seattle, WA 98195, USA

**Keywords:** cardiomyopathy, pulmonary hypertension, mitral valve, valve repair

## Abstract

The combination of hypertrophic cardiomyopathy with outflow tract obstruction, severe pre-capillary and post-capillary pulmonary hypertension, and severe primary mitral regurgitation is rare and presents distinct management challenges. **Background and Clinical Significance**: Pulmonary hypertension is an independent predictor of all-cause mortality in patients with hypertrophic cardiomyopathy managed medically and often precludes patients from undergoing cardiopulmonary bypass due to increased surgical morbidity and mortality. In studies specifically evaluating surgical myectomy, however, survival is favorable in patients with moderate-to-severe pulmonary hypertension. **Case Presentation**: We present a case of a 74-year-old male with six months of dyspnea with minimal exertion. A diagnostic work-up with transthoracic echocardiogram showed asymmetric left ventricular hypertrophy, left ventricular outflow tract obstruction with a peak gradient of 200 mmHg, right ventricular systolic pressure of 99 mmHg, systolic anterior motion of the mitral valve and flail anterior mitral leaflet. The patient was evaluated by a multi-disciplinary team and underwent extended septal myectomy and mitral valve repair with significant improvement in functional capacity post-operatively. **Conclusions**: While pulmonary hypertension increases the risk of morbidity and mortality during cardiopulmonary bypass, moderate-to-severe pulmonary hypertension in hypertrophic cardiomyopathy with outflow tract obstruction is a unique indication for septal reduction therapy that may not be associated with higher surgical mortality.

## 1. Introduction and Clinical Significance

Hypertrophic cardiomyopathy (HCM) is a common heritable condition that, when associated with additional cardiopulmonary co-morbidities, can present challenges for both diagnosis and management. While HCM in and of itself is common, it is rare to see this clinical condition present with both severe pulmonary hypertension and primary mitral regurgitation (MR). As such, few studies exist to offer guidance on the medical and surgical management of all three of these clinical conditions at once.

In this case, we describe a man diagnosed with HCM with outflow tract obstruction who was also found to have severe pre-capillary and post-capillary pulmonary hypertension (cpc-PH) and severe primary mitral regurgitation due to anterior leaflet prolapse and partial flail. Our work aims to highlight the utility of surgical management in this patient’s care.

This paper is an extended version of our abstract presented at a national American College of Cardiology meeting (New Orleans, Louisiana, March 2023, poster 2552) [1].

## 2. Case Presentation

### 2.1. Patient Information

A 74-year-old male with hypertension, hyperlipidemia, diabetes mellitu and chronic kidney disease presented with six months of dyspnea with minimal exertion.

### 2.2. Clinical Findings

On presentation, he was afebrile, bradycardic at 51 beats per minute and normotensive at 108/68 mmHg. His oxygen saturation was 94% on ambient air. A physical exam was notable for a 3/6 systolic ejection murmur in the right upper sternal border and a 3/6 holosystolic murmur at the apex.

### 2.3. Diagnostic Assessment

Transthoracic echocardiogram (TTE) showed severe asymmetric left ventricular hypertrophy, preserved left ventricular ejection fraction at 65–70%, systolic anterior motion (SAM) of the mitral valve (Figure 1 and Appendix A), left ventricular outflow tract (LVOT) obstruction with a peak velocity of 7.1 m/s at rest and an eccentric, posteriorly directed mitral regurgitant jet (Figure 2). Pharmacological myocardial perfusion SPECT with gated imaging was obtained one year prior to presentation and showed normal myocardial perfusion, left ventricular volume and systolic function. Cardiac magnetic resonance imaging showed left ventricular hypertrophy with a maximal thickness of 1.7 cm at the mid-ventricular septum and late gadolinium enhancement in the lateral and inferior half of the left ventricle, the basal inferolateral wall, the apical lateral walls and the apical anterior wall (Figure 3a,b). These findings were consistent with a diagnosis of HCM with outflow tract obstruction; thus, the patient was started on beta blockers.

The patient had initial symptomatic improvement after the initiation of beta blockers but returned three months later with worsening dyspnea. Repeat TTE showed the resolution of SAM and LVOT obstruction, indicating that the initial SAM was related to dynamic LVOT obstruction, flow acceleration and Venturi forces related to the hyperdynamic state. However, there was persistent posterior MR despite the resolution of SAM, indicating a SAM-independent cause of primary MR that may have been missed on transthoracic imaging (Figure 4 and Figure 5, Appendix A). Additionally, there was new right ventricular enlargement and an elevated pulmonary artery systolic pressure of 74 mmHg.

Given the concern for primary MR, a transesophageal echocardiogram (TEE) was performed for further assessment. The study showed prolapse, chordal rupture and partial flail of the A3 leaflet with an area of non-coaptation and severe MR (Figure 6, Appendix A). Quantification using the proximal isovelocity surface area method revealed an effective orifice area of 0.46 cm^2^ and a regurgitant volume of 80 milliliters with notable systolic pulmonary vein flow reversal, consistent with severe Carpentier class II MR from prolapse and flail (Figure 7, Appendix A) [2]. This highlights the role of TEE and 3D imaging in assessing for SAM-independent causes of primary MR in patients with HCM. A TTE alone can miss these anatomical findings, especially in medial (A3 or P3) or lateral (A1 or P1) mitral scallops.

Left and right heart catheterization showed moderate non-obstructive coronary disease and confirmed the presence of severe pulmonary hypertension (PH) with pulmonary arterial systolic, diastolic and mean pressures of 117/35/61 mmHg. The right atrial pressure was normal, at 4 mmHg, and pulmonary capillary wedge pressure (PCWP) was elevated to 25 mmHg with notable V-waves. The transpulmonary pressure gradient was calculated to be 36 mmHg. The cardiac index was calculated to be 2.6 L/min/m^2^ using the Fick equation, and pulmonary vascular resistance (PVR) was elevated to 6.9 Wood units. On a six-minute walk test, the patient walked 611 feet with an oxygen desaturation of 88%.

The initial work-up for PH revealed a severe reduction in diffusing capacity for carbon monoxide but no significant obstruction on pulmonary function testing and no evidence of chronic thromboembolic disease on a ventilation/perfusion lung scan. Laboratory tests for rheumatoid arthritis, systemic sclerosis and HIV were negative. In the setting of an elevated PCWP and transpulmonary pressure gradient greater than 12 mmHg, the suspected etiology for the patient’s severe PH was thought to be due to left sided heart disease and was further complicated by a pre-capillary component secondary to chronic pulmonary vascular remodeling.

In summary, the patient was diagnosed with HCM with outflow tract obstruction and concomitant severe MR secondary to mitral valve prolapse and flail, as well as severe cpc-PH.

### 2.4. Therapeutic Intervention

The patient was admitted to the intensive care unit for invasive hemodynamic monitoring with a Swan-Ganz catheter in place. Pulmonology was consulted for the treatment of cpc-PH in the setting of left heart disease, and the patient was initiated on sildenafil as some data have shown improvements in hemodynamics and exercise capacity with PDE5 inhibitors in this population [3]. He was initiated on 20 mg of sildenafil, which was later up-titrated to 40 mg. Although his pulmonary pressures improved to 78/12 mmHg (mean 34 mmHg) after 40 mg of sildenafil, he developed large left atrial V-waves and his systemic systolic blood pressure fell to 80 mmHg. He was subsequently trialed on 12.5 mg of sildenafil with an improvement in his hemodynamics (pulmonary pressures 88/56/66 mmHg, PCWP 23 mmHg, systolic blood pressure 120 mmHg). By hospital day four, he exhibited significant symptomatic improvement with a regimen of sildenafil 12.5 mg administered three times a day.

### 2.5. Follow-Up and Outcomes

The patient was evaluated via cardiothoracic surgery and underwent extended septal myectomy and complex mitral valve repair with chordae reconstruction, A3/P3 plication and annuloplasty ring insertion.

Pre-incision hemodynamics with isoproterenol challenge showed a peak LVOT gradient of 27 mmHg and a velocity of 1.6 m/s as well as a prominent septal knuckle under the aortic valve leaflets with progression towards the apex. A 1–1.5 cm trough was created longitudinally in the LVOT septal muscle, and a significant reduction was obtained using digital palpation. The myectomy resection line was extended to the mitral annulus medially and laterally, with the total excision measuring approximately 1.5 cm deep × 3 cm long × 4 cm wide. The pathology of this specimen under Gomori Trichrome stain showed a myocardium with cardiomyocyte disarray, hypertrophy and vacuolization, as well as patchy interstitial and endocardial fibrosis.

The mitral valve was examined intra-operatively and was noted to have thickened leaflets and an anterior ruptured chord at A3, rendering it flail. The leaflet was repaired with one neochord secured at the base of the anterolateral papillary muscle head. The anterior leaflet was measured from commissure to commissure, and a 34 mm Medtronic Simuform rigid posterior and semiflexible anterior segment was implanted. Hydrostatic testing demonstrated a leak through a small area of prolapse near A3 P3, which was repaired with a commissural plication stitch. Post-operative TEE with isoproterenol challenge showed no evidence of SAM or flow acceleration in the LVOT and trace MR. The post-surgical pulmonary artery systolic pressure was 27 mmHg immediately after intervention and increased to 60 mmHg in one month. The TTE obtained five months after intervention showed no LVOT obstruction, mild residual MR and a mitral inflow gradient of 6 mmHg. The patient’s six-minute walk test distance improved to 741 feet with the lowest oxygen saturation recorded at 94%. Right heart catheterization performed five months post-operatively showed pulmonary arterial systolic, diastolic and mean pressures of 78/27/47 mmHg, PCWP 18 mmH, PVR 7.8 Woods units and a Fick cardiac index of 1.99 L/min/m^2^. Compared to the pre-operative invasive assessment, the mean pulmonary artery pressure decreased; however, the cardiac output also decreased, resulting in similar and slightly higher post-operative PVR.

## 3. Discussion

HCM is frequently accompanied by abnormalities of the mitral valve, including leaflet elongation, anteriorly displaced and bifid papillary muscles and abnormal chordae attachment. These abnormalities, along with Venturi forces from flow acceleration across the LVOT, make the anterior leaflet prone to SAM with resultant LVOT obstruction [4,5]. SAM results in mitral leaflet malcoaptation and an eccentric, posteriorly directed mitral regurgitant jet, known as SAM-dependent MR. Typically, the relief of LVOT obstruction with either medical therapy or surgical intervention resolves MR without primary intervention on the mitral valve. However, HCM patients can also have SAM-independent MR due to other pathologies, such as prolapse, chordal rupture and leaflet flail. Given the risk of persistent MR after myectomy due to intrinsic mitral valve pathology, the American and European guidelines recommend patients undergo concomitant repair of intrinsic mitral valve disease [6,7,8,9].

Whether mitral valve disease is primary or secondary in this patient population remains controversial. Maron et al. evaluated 172 patients with preclinical HCM and found that both anterior and posterior leaflet elongation were present independent of myocardial hypertrophy, suggesting that this may represent a primary phenotypic expression of the disease [10,11]. The primary mechanism for mitral valve leaflet growth in HCM is poorly understood, with proposed etiologies including paracrine messaging from hypertrophic myocardium via periostin and mechanically induced growth in the setting of turbulent flow in the LVOT [10,12]. Interestingly, our patient had an elongated anterior mitral valve leaflet at 2.7 cm as well as bifid anterolateral and posteromedial papillary muscles. After the resolution of SAM and LVOT obstruction on medical therapy, our patient was noted to have persistent posteriorly directed MR. This raised the concern for a primary mitral valve pathology, specifically anterior mitral leaflet prolapse. Mitral valve chordae rupture in HCM is rare, with an estimated prevalence ranging between 1% and 5.4% [13,14]. While the pathogenesis of chordae rupture remains unclear, it is presumed that drag forces in the LVOT disrupt the structural integrity of elongated leaflets during SAM.

PH has been reported in 11–50% of patients with HCM, with moderate-to-severe PH seen in 12.5–18% of patients and up to 11% of patients meeting criteria for pre-capillary PH [15,16,17,18]. The presence of PH is considered an independent risk factor for HCM-related mortality. In such patients, pulmonary hypertension is associated with older age, higher BMI, greater wall thickness and left atrial volume, and female gender. Atrial fibrillation and moderate-to-severe mitral regurgitation are independent risk factors [16,17,18]. In a large study of 1570 HCM patients at the Mayo Clinic, 11.8% with HCM with outflow tract obstruction had moderate-to-severe PH. In those treated with medical therapy alone, pulmonary hypertension was associated with increased mortality. However, in those who underwent septal reduction therapy, either surgical myectomy or alcohol septal ablation, there was no difference in the post-operative survival between those with and without PH, further demonstrating the benefit for septal reduction therapy in these patients [19]. The decision between surgical myectomy and alcohol septal ablation needs to be individualized based on patient and anatomic characteristics.

Typically, when present prior to cardiopulmonary bypass, PH is associated with increased operative morbidity and mortality due to the risk of right ventricular failure [20,21]. Therefore, in patients with HCM with outflow tract obstruction and moderate-to-severe PH in need of septal reduction therapy, the decision may favor alcohol septal ablation. However, in our patient with concomitant primary mitral regurgitation in need of surgical repair, the decision to proceed with open heart surgery had to be considered. In a retrospective study of 306 patients with HCM who underwent surgical myectomy, 17% had moderate-to-severe PH. The group with moderate-to-severe PH were older, were more likely female, were more likely to have atrial fibrillation, and had higher natriuretic peptides and worse aerobic capacity as assessed on cardiopulmonary stress testing prior to surgery. Post myectomy, the 3-year survival rate was 97.5%, and having pre-operative PH did not predict survival, contradicting preconceptions of high operative morbidity. Additionally, the post-operative improvement in PH, measured based on a reduction in right ventricular systolic pressure, was best in patients with moderate-to-severe pre-operative PH [21]. Based on these data, the decision was made to pursue a surgical approach in this patient.

Medical therapy for cpc-PH in patients with HCM with outflow tract obstruction poses a unique conundrum, as phosphodiesterase inhibitors, commonly used to treat pre-capillary PH, can exacerbate pressure gradients in the LVOT. However, the off-label use of sildenafil in post-capillary PH is valued for its immediate ability to lower pulmonary systolic pressures and increase cardiac output [22]. While there are no specific data on the use of pulmonary vasodilators in patients with HCM with outflow tract obstruction, data from the SIOVAC trial showed worse clinical outcomes in patients with residual PH after surgically corrected valvular disease treated with sildenafil. Follow-up after 6 months in SIOVAC showed that patients on sildenafil were two times more likely to be admitted for acute decompensated heart failure requiring intravenous diuresis [22]. The decision was made to defer sildenafil use in our patient after surgery based on data from the SIOVAC trial.

## 4. Conclusions

Patients with HCM, severe PH and mitral valve disease should undergo a multidisciplinary evaluation. While PH increases the risk of morbidity and mortality during cardiopulmonary bypass, moderate-to-severe PH in HCM with outflow tract obstruction is a unique indication for septal reduction therapy that may not be associated with higher surgical mortality. The pathogenesis of MR in HCM is complex, with data suggesting that leaflet elongation may represent a primary phenotypic expression of the disease. Mitral valve chordae rupture in HCM, however, is a rare occurrence. Currently, there are no data to guide the medical management of persistent severe cpc-PH after septal reduction therapy in HCM. However, chronic use of sildenafil in patients with persistent PH after surgically corrected valvular disease doubles the risk for heart failure admission and all-cause mortality. Overall, this case describes the successful surgical management of HCM with outflow tract obstruction and severe MR secondary to a partial flail leaflet, with severe cpc-PH. Additional studies are needed to better understand the optimal medical and surgical treatment of this specific patient population.

## Figures and Tables

**Figure 1 reports-07-00116-f001:**
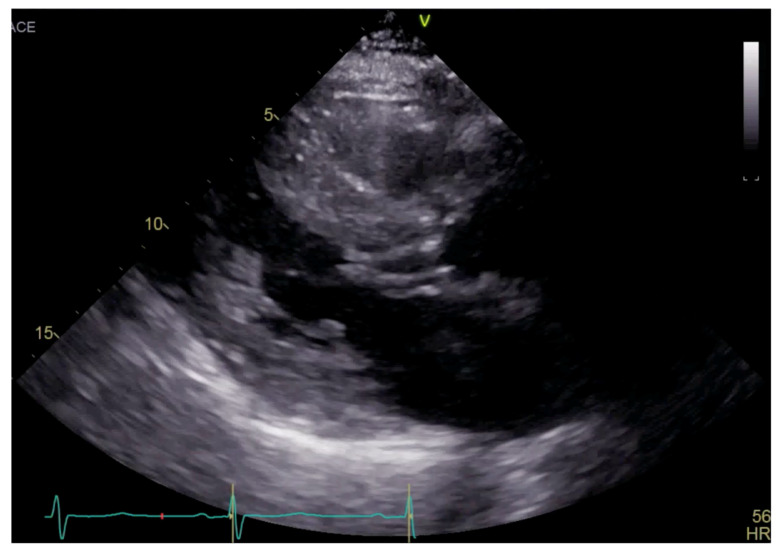
Parasternal long axis view on TTE showing left ventricular hypertrophy and SAM.

**Figure 2 reports-07-00116-f002:**
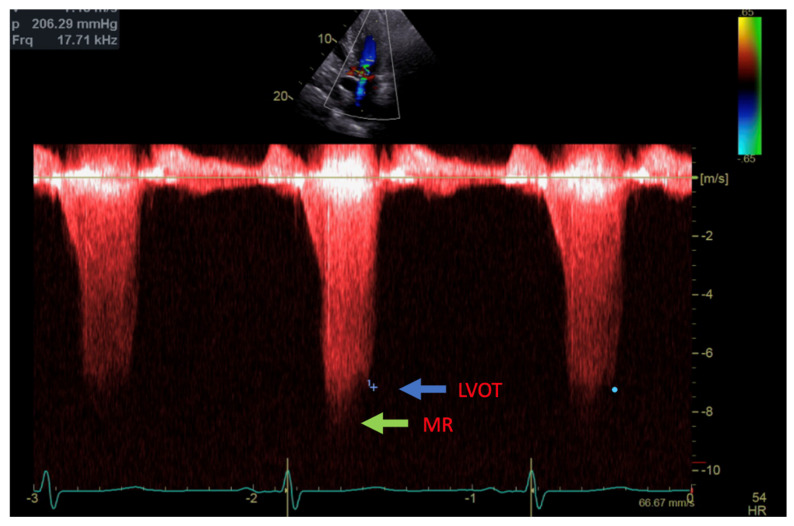
Continuous Doppler with two jets showing MR (green arrow) and LVOT gradient (blue arrow).

**Figure 3 reports-07-00116-f003:**
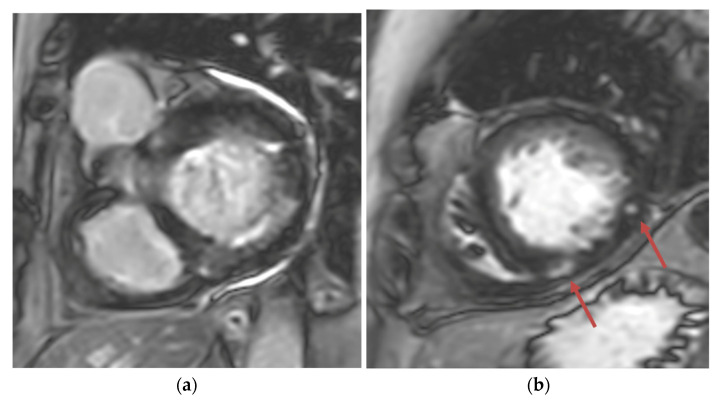
(**a**) Short-axis phase-sensitive inversion recovery showing patchy subendocardial late gadolinium enhancement in basal inferolateral wall; (**b**) Short-axis phase-sensitive inversion recovery showing patchy intramyocardial late gadolinium enhancement (red arrows).

**Figure 4 reports-07-00116-f004:**
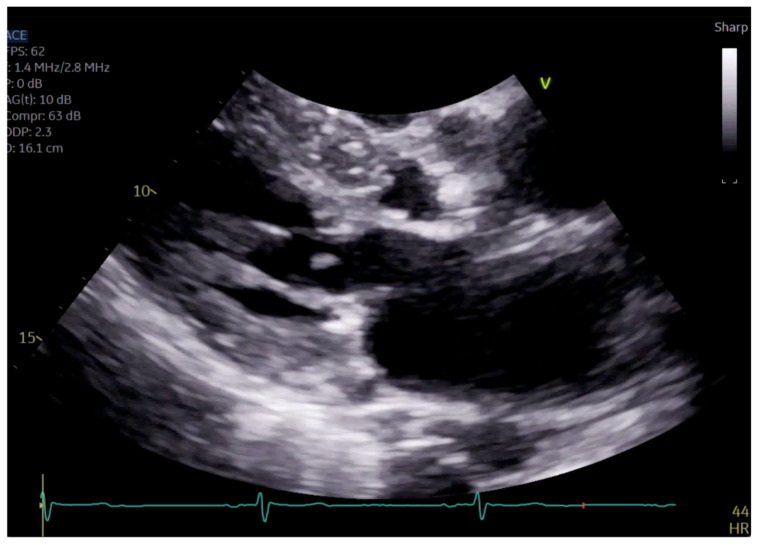
Close up of mitral valve on parasternal long axis view.

**Figure 5 reports-07-00116-f005:**
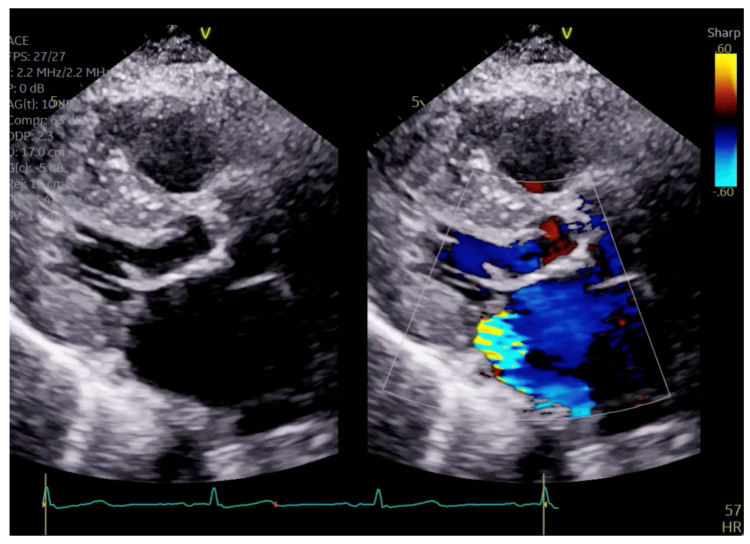
Parasternal long axis view showing posteriorly directed MR.

**Figure 6 reports-07-00116-f006:**
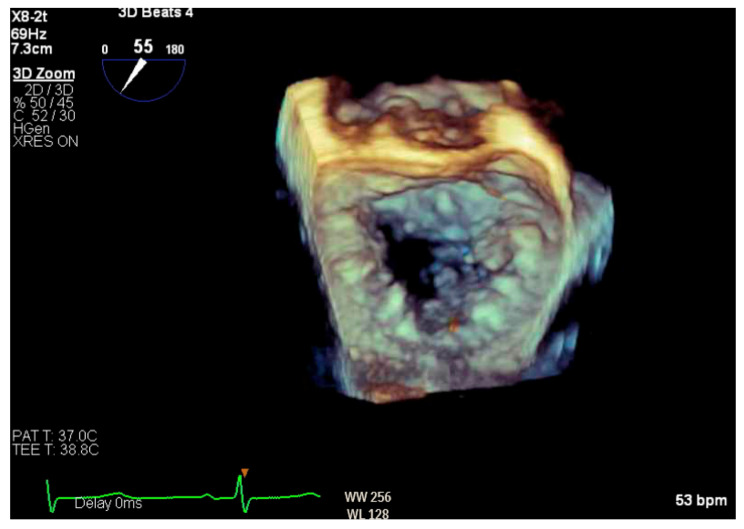
Three-dimensional reconstruction of mitral valve showing anterior leaflet prolapse and flail.

**Figure 7 reports-07-00116-f007:**
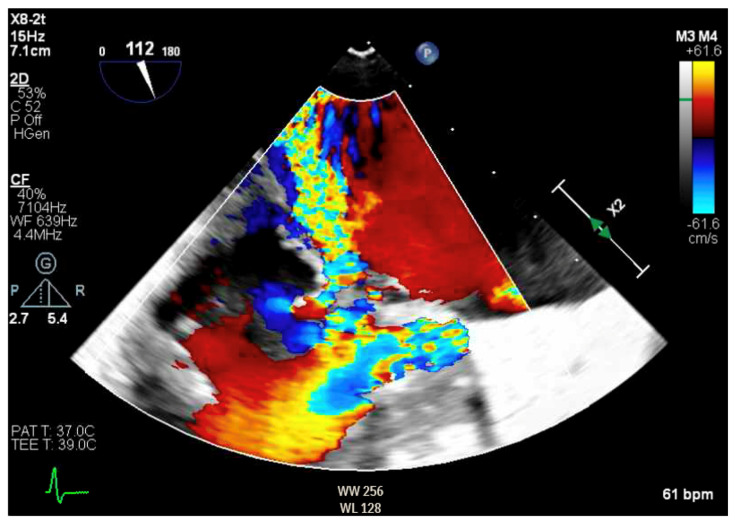
TEE demonstrating posteriorly directed MR.

## Data Availability

The original contributions presented in this study are included in the article/Appendix A. Further inquiries can be directed to the corresponding author.

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
