# Peer review of "A Triple Threat: A Case Report Detailing Surgical Management for Hypertrophic Cardiomyopathy, Flail Mitral Valve and Severe Pulmonary Hypertension†"

_reports, 2024, doi:10.3390/reports7040116_

Round 1
Reviewer 1 Report
Comments and Suggestions for Authors
It is not clear what was the initial SAM syndrome on the background of hypertrophy of the basal part of the interventricular septum or SAM appeared on the background of chordal detachment and changes in the coaptation zone of the flaps? Accordingly, it is impossible to say with certainty what caused the pulmonary hypertension. In this regard, I consider the argument about the prescription of sildanofil to be a stretch. No indication of how much myocardium was dissected, were there abnormal structures of the subvalvular apparatus that caused the SAM syndrome? There is no morphologic study of the dissected fragment of the interventricular septum, so it is difficult to speculate on the etiology of the lesion. There is no description of the mitral valve leaflets. What kind of support ring was implanted (size, stiffness, closed or C-shaped?) The bibliography does not include publications by A. Carpanteer.
Author Response
Comment 1:
It is not clear what was the initial SAM syndrome on the background of hypertrophy of the basal part of the interventricular septum or SAM appeared on the background of chordal detachment and changes in the coaptation zone of the flaps?
Response 1:
We appreciate the reviewer for asking this question and agree with the need to clarify the mechanism for SAM in this patient. We suspect that the primary mechanism was due to Venturi forces from flow acceleration across the LVOT given the resolution of SAM after initiation of beta blockade. We then suspected a SAM-independent cause of MR in this patient, prompting further evaluation with TEE and 3D imaging and the diagnosis of Carpentier Class II MR due to prolapse and flail. We have edited the manuscript to reflect this in lines 68-86.
Comment 2:
Accordingly, it is impossible to say with certainty what caused the pulmonary hypertension. In this regard, I consider the argument about the prescription of sildenafil to be a stretch.
Response 2:
We appreciate the reviewer for this comment and agree. Given that other causes of pulmonary hypertension were reasonably excluded, it was felt that the cause of pulmonary hypertension was mixed pre-capillary and post capillary pulmonary hypertension from combined left sided heart disease (valvular disease and diastolic dysfunction from hypertrophic cardiomyopathy). In consultation with our Pulmonology colleagues, sildenafil was chosen for initial management as there have been a few small studies that have shown improvement in PA pressures and symptoms in similar patients treated with PDE inhibitors, while other classes of PH therapies have not been shown to have improvement in this subgroup of patients. The hope was to improve hemodynamics prior to surgery. To help clarify our clinical reasoning for this diagnosis, we have added additional insight on our thoughts in lines 84-88. To help support our clinical reasoning for the use of sildenafil, we have included an additional citation and brief description in lines 151-154.
Reference:
- Lteif C.; Ataya A.; Duarte J.D. Therapeutic challenges and emerging treatment targets for pulmonary hypertension in left heart disease. Am. Heart. Assoc. 2021, 10, e020633. https://doi.org/10.1161/JAHA.120.020633.
Comment 3:
No indication of how much myocardium was dissected, were there abnormal structures of the subvalvular apparatus that caused the SAM syndrome?
Response 3:
We appreciate the reviewer for asking this question. We have clarified how much of the myocardium was dissected and the abnormal structures visualized intra-operatively in lines 166-172.
Comment 4:
There is no morphologic study of the dissected fragment of the interventricular septum, so it is difficult to speculate on the etiology of the lesion. There is no description of the mitral valve leaflets.
Response 4:
We appreciate the reviewer for this comment. We have included the pathology results from the dissected fragment of the interventricular septum in lines 172-174. We have also included the intra-operative description of the mitral valve in lines 175-176.
Comment 5:
What kind of support ring was implanted (size, stiffness, closed or C-shaped?)
Response 5:
We appreciate the reviewer for this question and have included clarification on the type of support rings used in lines 177-181. We additionally detailed the neochord insertion for repair of A3 flail in lines 176-177.
Comment 6:
The bibliography does not include publications by A. Carpentier.
Response 6:
We appreciate the reviewer for this comment. The patient was found to have Carpentier Class II MR from prolapse and flail. We included this information in lines 82-84 and added a reference by A. Carpentier to the bibliography.
Reference:
- Carpentier A. Cardiac valve surgery--the "French correction". J. Thorac. Cardiovasc. Surg. 1983, 86, 323–337. https://doi.org/10.1016/S0022-5223(19)39144-5.
Reviewer 2 Report
Comments and Suggestions for Authors
Title: A Triple Threat: Surgical Management for HCM, Flail Mitral Valve and Severe Pulmonary Hypertension.
Reviewer Summary: It is quite uncommon and poses unique therapeutic issues when HCM with outflow tract obstruction, significant pre- and post-capillary pulmonary hypertension (PH), and severe primary mitral regurgitation coexist. In patients with medically treated HCM, PH is an independent predictor of all-cause mortality and often prevents patients from having a cardiopulmonary bypass because of the higher risk of surgical morbidity and mortality. However, mortality rate is less in patients with moderate-to-severe PH in trials that specifically evaluate surgical myectomy. Here authors describing a case report pertaining to a 74-year-old male who has had mild exertion-induced dyspnea for 6 months. Asymmetric left ventricular hypertrophy, ventricular outflow tract obstruction with a peak gradient of 200 mmHg, right ventricular systolic pressure of 99 mmHg, systolic anterior motion of the mitral valve, and flail anterior mitral leaflet were all observed during the diagnostic work-up using a transthoracic echo. After extensive septal myectomy and mitral valve replacement, the patient's functional capacity dramatically improved after being assessed by a multidisciplinary team. In conclusion moderate-to-severe pulmonary hypertension in HCM with outflow tract obstruction is a special indication for septal myectomy that might not be associated with increased surgical mortality, even though PH raises the risk of morbidity and mortality during cardiopulmonary bypass.
Points to think about:
1. Was stress test done to the patient to evaluate the heart wellness?
2. Authors mentioned that PH increases the risk of morbidity and mortality during cardiopulmonary bypass, moderate-to-severe pulmonary hypertension in HCM with outflow tract obstruction is a unique indication for septal reduction therapy that may not be associated with higher surgical death. Are these same results shown in any other patients? Or these results are true in only one patient. In other words, can we generalize these results to other people?
3. Results from a single individual may not be applicable to other populations due to variations in individual characteristics and circumstances.
4. Would you see the same results in other age group of populations?
5. Are there any controls in this study?
6. Would HCM with moderate PH decreases the mortality rate in women as well?
7. What is the genetic cause of HCM in this patient?
8. Authors mentioned that PH increases the risk of mortality during cardiopulmonary bypass. What is the contribution of diabetes mellitus and chronic kidney disease in this patient?
9. HCM with moderate PH combination damage left ventricle performance. How does it correlate with decreased mortality?
Author Response
Comment 1:
Was stress test done to the patient to evaluate the heart wellness?
Response 1:
We appreciate the reviewer for this question and have included the patient’s normal stress test one year prior to presentation in lines 59-61. Additional stress testing was not performed as the patient was too symptomatic to undergo further evaluation and cardiac catheterization was pursued instead.
Comment 2:
Authors mentioned that PH increases the risk of morbidity and mortality during cardiopulmonary bypass, moderate-to-severe pulmonary hypertension in HCM with outflow tract obstruction is a unique indication for septal reduction therapy that may not be associated with higher surgical death. Are these same results shown in any other patients? Or these results are true in only one patient. In other words, can we generalize these results to other people?
Response 2:
We appreciate the reviewer for these questions. Chronic thromboembolic pulmonary hypertension is another clinical entity associated with severe PH that is treated surgically with pulmonary endarterectomy. At experienced medical centers with high volumes of this procedure, along with careful patient selection through multi-disciplinary team evaluation, mortality rates are estimated to be less than 5% for this procedure. We believe that these results highlight the need for more research on the intra-operative risk of severe PH in patients who may benefit from surgical intervention to reverse the primary causative condition, though we hesitate to say that these results can be generalized to others without additional data.
Reference:
- Jenkins D. Pulmonary endarterectomy: the potentially curative treatment for patients with chronic thromboembolic pulmonary hypertension. Respir. Rev. 2015, 136. https://doi.org/10.1183/16000617.00000815.
Comment 3:
Results from a single individual may not be applicable to other populations due to variations in individual characteristics and circumstances.
Response 3:
We agree with the reviewer’s comment. We have updated the discussion section to more clearly outline findings from a larger retrospective cohort of 306 patients with HCM with obstruction treated with myectomy. In this group, 51 patients had moderate to severe pulmonary, similar to our patient.
Comment 4:
Would you see the same results in other age group of populations?
Response 4:
We appreciate the reviewer’s question. We note that in the study looking at those who underwent surgical myectomy, the patients with moderate to severe pulmonary hypertension were older that those without pulmonary hypertension. The median age was 51-year- old without PH versus 63-years-old (48-72 years-old) with moderate to severe pulmonary hypertension. Therefore, we think that our patient fits nicely into this study.
Reference:
- Geske JB.; Konecny T.; Ommen SR.; et al. Surgical myectomy improves pulmonary hypertension in obstructive hypertrophic cardiomyopathy. Eur. Heart. J. 2014, 35, 2032–2039. https://doi.org/1093/eurheartj/eht537.
Comment 5:
Are there any controls in this study?
Response 5:
We appreciate the reviewer’s question. We have updated the discussion to indicate that the above mentioned study (Geske at al), which evaluated patients treated with myectomy, compared those without pulmonary hypertension to those with moderate to severe pulmonary hypertension.
Comment 6:
Would HCM with moderate PH decreases the mortality rate in women as well?
Response 6:
We appreciate the reviewer’s question. We have updated the discussion to indicate that in this population, patients with moderate to severe pulmonary hypertension are more likely to be female. While there was no covariate analysis based on outcomes and gender, we feel that females are well represented in this study.
Reference:
- Geske JB.; Konecny T.; Ommen SR.; et al. Surgical myectomy improves pulmonary hypertension in obstructive hypertrophic cardiomyopathy. Eur. Heart. J. 2014, 35, 2032–2039. https://doi.org/1093/eurheartj/eht537.
Comment 7:
What is the genetic cause of HCM in this patient?
Response 7:
We appreciate the reviewer’s question. We did not pursue genetic testing to determine the cause of HCM in this patient as he does not have any siblings or children and did not wish to pursue this testing.
Comment 8:
Authors mentioned that PH increases the risk of mortality during cardiopulmonary bypass. What is the contribution of diabetes mellitus and chronic kidney disease in this patient?
Response 8:
The relationship between diabetes mellitus and cardiopulmonary bypass has been studied in long-term outcomes after coronary artery bypass grafting. Diabetes was found to be a risk factor for early and late mortality after revascularization (1). For general peri-operative management, inadequately controlled diabetes is associated with poor wound healing. One week prior to surgery, the patient’s diabetes was well controlled with a hemoglobin A1c of 7.1%.
The contribution of chronic kidney disease on mortality in cardiopulmonary bypass is not well described. There is literature on the increased risk of developing acute kidney injury after cardiopulmonary bypass, which would be further increased by this patient’s baseline chronic kidney disease (2).
References:
1. van Straten A.H.; Soliman Hamad MA.; van Zundert A.A.; et al. Diabetes and survival after coronary artery bypass grafting: comparison with an age- and sex-matched population. J. Cardiothorac. Surg. 2010, 37, 1068-1074. https://doi.org/10.1016/j.ejcts.2009.11.042.
2. Milne B.; Gilbey T.; De Somer F.; Kunst G. Adverse renal effects associated with cardiopulmonary bypass. Perfusion. 2024, 39, 452-468. https://doi.org/10.1177/02676591231157055.
Comment 9:
HCM with moderate PH combination damage left ventricle performance. How does it correlate with decreased mortality?
Response 9:
We appreciate the reviewer’s question. Our intention was to highlight that, contrary to known increased operative risks associated with pulmonary hypertension, Ong et al found that pre-operative PH did not predict morbidity and mortality after septal reduction therapy in patients with HCM. We did not mean to necessarily highlight any evidence decreased mortality from an intervention.
To clarify these intentions, we edited the discussion to include additional text from line 221-260.
Reference:
- Ong KC.; Geske JB.; Hebl VB.; et al. Pulmonary hypertension is associated with worse survival in hypertrophic cardiomyopathy. Heart. J. Cardiovasc. Imaging. 2016, 17, 604–610. https://doi.org/10.1093/ehjci/jew024.